# Development and Characterization of Inducible Astrocyte-Specific Aromatase Knockout Mice

**DOI:** 10.3390/biology12040621

**Published:** 2023-04-19

**Authors:** Jing Wang, Uday P. Pratap, Yujiao Lu, Gangadhara R. Sareddy, Rajeshwar R. Tekmal, Ratna K. Vadlamudi, Darrell W. Brann

**Affiliations:** 1Department of Neurosurgery, Medical College of Georgia, Augusta University, Augusta, GA 30912, USA; 2Department of Obstetrics and Gynecology, University of Texas Health, San Antonio, TX 78229, USA

**Keywords:** estradiol, neuroestrogen, neurosteroid, aromatase, cerebral ischemia, astrocyte, microglia

## Abstract

**Simple Summary:**

Recent work has shown that the steroid hormone17β-estradiol (E_2_) is also produced in the brain in both neurons and astrocytes. The current animal models for depleting E_2_, specifically in astrocytes, are non-inducible, which can provide a developmental confound. It would be advantageous to have an inducible knockout model where the E_2_ depletion in astrocytes could be performed in adult animals, so as to avoid this confound. Thus, in the current study, we created an inducible knockout mouse model to deplete the E_2_ specifically in the astrocytes of adult mice. The characterization of the inducible knockout mice confirmed that aromatase and E_2_ were depleted specifically in the astrocytes of the adult mice. The study further revealed that astrocyte-derived E_2_ had a significant role in protecting the brain from cerebral ischemia (stroke) and regulated both the astrocyte and microglia activation after a cerebral ischemia.

**Abstract:**

17β-estradiol (E2) is produced in the brain as a neurosteroid, in addition to being an endocrine signal in the periphery. The current animal models for studying brain-derived E_2_ include global and conditional non-inducible knockout mouse models. The aim of this study was to develop a tamoxifen (TMX)-inducible astrocyte-specific aromatase knockout mouse line (GFAP-ARO-iKO mice) to specifically deplete the E_2_ synthesis enzymes and aromatase in astrocytes after their development in adult mice. The characterization of the GFAP-ARO-iKO mice revealed a specific and robust depletion in the aromatase expressions of their astrocytes and a significant decrease in their hippocampal E_2_ levels after a GCI. The GFAP-ARO-iKO animals were alive and fertile and had a normal general brain anatomy, with a normal astrocyte shape, intensity, and distribution. In the hippocampus, after a GCI, the GFAP-ARO-iKO animals showed a major deficiency in their reactive astrogliosis, a dramatically increased neuronal loss, and increased microglial activation. These findings indicate that astrocyte-derived E_2_ (ADE_2_) regulates the ischemic induction of reactive astrogliosis and microglial activation and is neuroprotective in the ischemic brain. The GFAP-ARO-iKO mouse models thus provide an important new model to help elucidate the roles and functions of ADE_2_ in the brain.

## 1. Introduction

The biosynthetic enzyme aromatase produces estrogens from androgen precursor steroids [1,2,3]. 17β-Estradiol (E_2_) is the most potent estrogen produced in the body and acts through three major receptors—estrogen receptor-alpha (ER-α), estrogen receptor-beta (ER-β), and G-coupled protein estrogen receptor 1 (GPER1) [4,5,6,7]. While aromatase is well known to be highly expressed in the ovaries of females, it has also been shown to be expressed in the neurons and astrocytes in the brains of both males and females of many species, and can generate significant levels of E_2_ [8,9,10,11,12,13]. Interestingly, aromatase is constitutively expressed in neurons [13,14,15,16], while its expression is inducible in astrocytes [17,18,19]. Stressors such as brain injury and ischemia have been shown to strongly induce both the expression and activity of this aromatase in astrocytes and to markedly increase the E_2_ levels in the brain [17,18,19,20,21]. The aromatase expression in rodent neurons has been found to be highest in the amygdala, hippocampus, hypothalamus, and cortex of males and females [22]. Similarly, positron emission tomography (PET) imaging has confirmed the highest aromatase expression in the human brain to be in the amygdala, preoptic area, thalamus, hippocampus cerebellum, white matter, and cortex in both males and females [23]. Additional studies have demonstrated that the aromatase expression in neurons is localized in axons, dendrites, synapses, and presynaptic terminals [16,24]. Aromatase activity is regulated both transcriptionally and post-translationally. The rapid regulation of aromatase activity can occur through calcium-dependent phosphorylation [25]. In addition, the neurotransmitter glutamate has been shown to regulate the aromatase activity in the brains of many species [26]. The roles of neuron-derived E_2_ (NDE_2_) and astrocyte-derived E_2_ (ADE_2_) in the brain have been explored using various approaches, including the administration of aromatase inhibitors [8,27,28,29,30,31,32] and antisense oligonucleotides to inhibit or knockdown the aromatase centrally [12]. Global aromatase knockout mice [33] and aromatase knockout mouse models that are specific to astrocytes and neurons, respectively, have been created by our team [21,34]. The available evidence to date indicates that NDE_2_ regulates neurotransmission, synaptic plasticity, cognitive function, injury-induced reactive gliosis, neuroinflammation, and neuroprotection [8,32]. With respect to ADE_2_, recent work from our group, using a non-inducible GFAP-cre-driven knockout of the aromatase in mice (GFAP-ARO-KO mice), has implicated a role for ADE_2_ in the regulation of glia activation, anti-inflammatory effects, and neuroprotection after a global cerebral ischemia (GCI) in mice [21]. However, in addition to being expressed in astrocytes, GFAP is also expressed in the progenitor cells that produce non-astrocyte cells during development [35], which could provide a potential confound. The aim of the current study was to develop a tamoxifen (TMX)-inducible astrocyte-specific aromatase knockout mouse line (GFAP-ARO-iKO mice) that allows for the specific depletion of the aromatase in astrocytes after their development in adult mice, and it phenotypically characterized the mice following a GCI. The results revealed the successful depletion of the aromatase, specifically in the astrocytes of adult GFAP-ARO-iKO mice, and a robust phenotype after a GCI.

## 2. Materials and Methods

### 2.1. Generation of a Mouse Model of Inducible Astrocyte-Specific Aromatase Knockout 

We created the *Cyp19a1* Cre/LoxP conditional KO mice with an inducible GFAP promoter to conditionally inactivate the aromatase in their astrocytes. GFAP-Cre/ERT2 (Jackson lab, Bar Harbor, ME, USA. stock no. 012849) mice were crossed with *Cyp19a1 flox/flox* animals, which had LoxP sites surrounding exons 10 and 11 of the *Cyp19a1* gene. Using the methods previously reported, *Cyp19a1* flox/flox (FLOX) mice were produced [34]. We confirmed the genotyping of the animals by a genomic DNA PCR, using the following primers: FLOX-forward: 5′-GCCATATTCCTGCAACAGTTTATTTGAGG-3′; FLOX-reverse: 5′-GTAAAACCATTCCTGGAAAATTCATAACAACC-3′; CreERT2-forward: 5′-GGCCAGGCTGTTCTTCTTAG-3′; CreERT2-reverse: 5′-ATACCGGAGATCATGCAAGC-3′; Internal positive control forward: 5′-CTAGGCCACAGAATTGAAAGATCT-3′; and Internal positive control reverse: 5′-GTAGGTGGAAATTCTAGCATCATCC-3′. The mice with the genotype of *Cyp19a1flox/flox*; *GFAPCreERT2* were identified as the GFAP-ARO-iKO mice. All the animal experiments were approved and authorized by the Institutional Animal Care and Use Committees of Augusta University and UT Health San Antonio.

### 2.2. Induction of Aromatase Deletion in the GFAP-ARO-iKO Mice 

To induce an aromatase deletion in the astrocytes, we administered tamoxifen (TMX, 100 mg/kg) through intraperitoneal injections for 3–5 consecutive days. The 3-day TMX injections were administered in the initial experiment to confirm the down-regulation of the aromatase in the astrocytes after a GCI using purified astrocytes. The animals from all the other experiments received injections of the TMX for 5 consecutive days. A 3-week washout period was used in our studies before performing the experiments. We also included several controls, including GFAPCreERT2 mice, ARO-FLOX mice with TMX injections, and GFAP-ARO-iKO mice without TMX injections. 

### 2.3. Global Cerebral Ischemia

A two-vessel occlusion global cerebral ischemia (GCI) was performed on male mice from the different experimental groups after the 3-week TMX wash-out period, as described previously [21]. Analgesia was given 16 h before and 0–16 h after surgery by providing carprofen (2 mg Rimadyl, formulated in a 5 g food tablet). The surgeries were performed on the anesthetized mice by giving 1–4% isoflurane, which was inhaled before and during the operation. For the GCI, a midline incision of the neck was made and both the common carotid arteries (CCAs) were exposed and encircled loosely with a 4-0 suture. Then, a GCI was induced by bilaterally blocking the CCAs for 30 min. The clips were taken off after the ischemia and a visual examination of the blood flow in the CCAs proved that reperfusion had occurred. A homoeothermic blanket kept the mice’s core body temperature at 37 °C before, during, and after the ischemia. The mice were kept in a warm chamber for five minutes after the reperfusion to keep their body temperatures at about 37°C before the incisions were closed. The CCAs were exposed without an occlusion for the animals in the sham groups. The mice that displayed dilated pupils during the surgery and lost the righting reflex for at least one minute following the wound closure were included in the subsequent studies. The mice that demonstrated signs of respiratory distress and failed to return to normal activity within the first 24 h were euthanized and excluded from the subsequent analysis. The initial number of the animals assigned to the sham groups was 6, and for the GCI groups, it was 8. In total, 10% of the animals were excluded due to surgery complications and the criteria described above. 

### 2.4. Tissue Sample Collection and Astrocyte Purification 

The mice were sacrificed under deep anesthesia 3 days after the GCI surgery, as this is a time of robust aromatase induction in astrocytes after a GCI [21]. Ice-cold 0.9% saline was used to perfuse the mice and their hemispheres were dissected. To verify the astrocyte-specific deletion of the aromatase in the GFAP-ARO-iKO mice, we purified the astrocytes from the mouse brains using magnetic associated cell sorting (MACS) with ACSC2 Microbeads. Using an adult brain dissociation kit, a single-cell solution was prepared (Miltenyi Biotec, Bergisch Gladbach, Germany; 130-107-677), as previously described [21]. Briefly, the purified astrocytes were generated by dissociating single cells through the MACS separation columns using ACSC2 Microbeads, as described by the manufacturer (Miltenyi Biotec, 130-093-634). In a previous study, we demonstrated that this method generates a highly purified astrocyte population, with >95% cells expressing the ACSC2 astrocyte cell marker [21]. For each isolation, four hemispheres were pooled in the study. At least 2 isolations were performed for each group.

### 2.5. Cell Lysates Preparation and Western Blotting 

The purified astrocytes from three animals were pooled to prepare the lysates for a Western blot analysis. A radioimmunoprecipitation assay buffer (150 mM NaCl, 1 mM EDTA, 1% Triton X-100, 1% sodium deoxycholate, 50 mM Tris-HCl at pH 7.4, and 0.1% SDS), supplemented with protease and phosphatase inhibitors, was used to extract the total proteins from the dissected mouse brain tissue (ThermoFisher Scientific, Waltham, MA, USA). A microplate BCA protein assay kit was used to determine the samples’ protein quantities (ThermoFisher Scientific). Sodium dodecyl-sulfate polyacrylamide gel electrophoresis (SDS-PAGE) was used to separate equal amounts of the proteins (20 or 40 g), which were then transferred to nitrocellulose membranes for a Western blot analysis, as previously described. [21]. The antibodies used for the Western blots included aromatase (rabbit, 1:1000; ThermoFisher Scientific, PA1-21398), GFAP (goat, 1:2000; Abcam, Waltham, MA, USA, ab53554), Vimentin (mouse, 1: 1000; Abcam, MA, USA, ab 20346), and GAPDH (mouse, 1:2000; Santa Cruz Biotechnology, Dallas, TX, USA, sc-32233). The blots were visualized using a LI-COR Odyssey imager after the blots were incubated with a secondary antibody. The intensity of each band was determined using ImageJ analysis software (Version 1.49; NIH, Bethesda, MD, USA). The normalization of band density to the corresponding loading controls was performed.

### 2.6. Immunohistochemistry (IHC) 

The brain sections were prepared as described previously [21]. Briefly, 3 days after the GCI surgery, the mice were sacrificed by an anesthetic overdose and perfused with cold saline, followed by 4% paraformaldehyde. Then, the brains were removed, 4% paraformaldehyde at 4 °C for 16 h was used for the post-fixation, and cryoprotection was performed with 30% sucrose. Next, 30 μm thick coronal sections were taken between −0.9 mm and −3.0 mm from the bregma. The brain sections were then blocked for 1 h at room temperature with 10% normal donkey serum, and then the primary antibodies were incubated overnight at 4 °C. The primary antibodies used in this study included aromatase (rabbit, 1:1000; ThermoFisher Scientific, PA1-21398), NeuN (rabbit, 1:1000; Millipore, Burlington, MA, USA, #ABN78), GFAP (goat, 1:2000; Abcam, MA, USA, ab53554), GFAP (mouse, 1:800; Millipore, MA, USA, MAB360), and Iba1 (goat, 1:400; Abcam, MA, USA, ab5076). Following a wash, the sections were incubated for 1 h at room temperature with the appropriate Alexa-Fluor-tagged secondary antibodies (ThermoFisher Scientific). Following that, the sections were stained with DAPI (4′,6-diamidino-2-phenylindole) and mounted in a Vectashield mounting media (Vector Laboratories, Newark, CA, USA). Images of the hippocampal CA1 region were acquired using a Zeiss 510 confocal microscope with a 40× Plan-Apochromat objective. The exposure time was kept constant for each marker in all the sections for the analysis. The intensity of the immunofluorescent staining in the hippocampal CA1 region was quantified with ImageJ software. The mean cell counts were calculated from three random microscopic fields at a 40× magnification for the quantifications of the neuron number and aromatase-expressing astrocytes. A total of 5–6 mice/groups were used for each analysis, and at least 3 sections from each animal were counted for each measurement.

### 2.7. Statistical Analyses

All of the data were examined using GraphPad Prism 8 Version 2 software (GraphPad, CA, USA). The data are displayed as means and SEM. A one-way ANOVA was used to compare the differences between the GFAP-ARO-iKO mice and other control groups, including the GFAPCreERT2 mice, ARO-FLOX mice with TMX injections, and GFAP-ARO-iKO mice without TMX injections. Two-way ANOVAs were used to compare the differences between the different groups in the sham conditions or after the GCI. When the ANOVA test was found to be significant, a post hoc Tukey’s test was conducted to make pairwise comparisons between the experimental groups. A value of *p* < 0.05 was considered to be statistically significant. 

## 3. Results

### Generation and Characterization of GFAP-ARO-iKO Mice

The GFAP-ARO-iKO mice were generated by crossing *GFAP-Cre/ERT2* mice with *Cyp19a1 flox/flox* (ARO-FLOX) mice, which contain LoxP sites flanking exons 10 and 11 of the Cyp19a1 gene (Figure 1A). The mice with the genotype of *Cyp19a1flox/flox;GFAPCreERT2* were identified as the GFAP-ARO-iKO mice (Figure 1B). 

The GFAPCreERT2 mice, ARO-FLOX mice with TMX injections, and GFAP-ARO-iKO mice without TMX injections served as controls for the study. To examine the efficacy of the TMX-induced aromatase knockdown in the astrocytes, purified astrocytes were isolated from the 3-month-old male sham or GCI reperfusion 3-day (R3d) mice, and a Western blot analysis was conducted using cell lysates from the astrocytes to detect the aromatase expressions. The results revealed, as expected, that the aromatase expression was low to undetectable in the astrocytes from the sham WT, FLOX, and GFAP-ARO-iKO animals. However, aromatase expression was strongly induced at R3d after the GCIs in the WT and FLOX astrocytes. Importantly, the expression of aromatase was robustly reduced (a 65–67.5% decrease) in the astrocytes from the TMX-injected GFAP-ARO-iKO mice, as compared to the FLOX group with TMX injections and the GFAP-ARO-iKO group without TMX injections, respectively (Figure 1C,D). 

We next examined several endpoints in the GFAP-ARO-iKO mice to further characterize the knockout efficiency and specificity. The GFAP-ARO-iKO and FLOX mice were used, along with the same controls, as described in Figure 1. The animals were examined at R3d after the GCI, as this is a time of robust induction of astrocyte aromatase and of hippocampal E_2_ levels after a GCI [21]. The GFAP-ARO-iKO mice were viable and fertile with normal gross brain structures. Co-immunostaining for the aromatase with GFAP in the hippocampal sections revealed that aromatase was absent in the astrocytes basally in the sham male FLOX and KO animals, while a normal aromatase expression was observed in the neurons (Figure 2A). 

As expected, a GCI increased the aromatase expression in the astrocytes of the male FLOX mice at R3d, and this effect was strongly attenuated in the GFAP-ARO-iKO mice (Figure 2A,B). The quantification of the number of cells that co-expressed GFAP and aromatase also demonstrated significant aromatase-expressing astrocytes at R3d in the FLOX mice, which was strongly attenuated in the GFAP-ARO-iKO mice at R3d (Figure 2C). In addition, the measurement of the hippocampal E_2_ levels, using a high-sensitivity ELISA, demonstrated a robust induction of the hippocampal E_2_ levels at R3d in the FLOX mice, and this effect was significantly reduced in the GFAP-ARO-iKO mice (Figure 2D). These in vivo findings further confirm the astrocyte-specific deletion of aromatase in the GFAP-ARO-iKO mice and show that this effect is associated with the significant diminishment of ischemia-induced hippocampal E_2_ elevation after a GCI. 

The GFAP-ARO-iKO mice exhibited a significant defect in reactive gliosis after the GCI. We next examined the reactive astrogliosis in the GFAP-ARO-iKO mice and control mice. Cell lysates taken from the isolated astrocytes at R3d after the GCI were subjected to a Western blot analysis to measure the expression of the reactive astrocyte markers, GFAP and vimentin. The results demonstrated a significant attenuation of the GCI-induced upregulation of GFAP and vimentin in the astrocytes from the GFAP-ARO-iKO mice with TMX injections, as compared to the GFAP-ARO-iKO non-TMX injected group and the FLOX+TMX injection group (Figure 3A–C). 

Likewise, an IHC performed on the sham group and R3d hippocampal sections demonstrated a significant decrease in the GFAP immunoreactive intensity in the GFAP-ARO-iKO mice at R3d, as compared to the control mice (Figure 2A,B). Additionally, it is notable that the astrocytes in the GFAP-ARO-iKO mouse hippocampus at R3d appear to exhibit significantly less hypertrophy and a reduced aromatase expression, as compared to the control mice (Figure 2A,C). These findings indicate that the GFAP-ARO-iKO mice have significantly reduced reactive astrogliosis after an ischemic brain injury. 

The GFAP-ARO-iKO mice exhibited a significantly enhanced neuronal loss and microglial activation after the GCI. Finally, the co-immunostaining of the neuron marker, NeuN, with the microglial activation marker, Iba1, demonstrated a significant decrease in the hippocampal CA1 NeuN+ cells in the GFAP-ARO-iKO mice at R3d, as compared to the FLOX mice, while in contrast, the Iba1 expression was significantly increased (Figure 4A–C). 

These findings indicate that the GFAP-ARO-iKO mice have a greater neuronal loss and increased microglial activation after a GCI, as compared to the control mice. Collectively, these findings confirm a key role of ADE_2_ in regulating astrocyte and microglial activation and exerting neuroprotection in the hippocampus following a GCI, and provide a new inducible mouse model for studying the roles and functions of ADE_2_ in the brain.

## 4. Discussion

In this study, we have successfully created an inducible model for the astrocyte-specific depletion of aromatase in adult mice, offering the field a crucial new animal model. Previous knockout models for aromatase have either been global in nature [33] or astrocyte-specific [21], and were non-inducible. While non-inducible GFAP-ARO-KO mice have provided important new insights into the various roles of ADE_2_ in the brain [21], the expression of GFAP in non-astrocytes during development could provide a potential confound. The development of GFAP-ARO-iKO mice allows for the circumvention of this potential confound by providing a mechanism for the astrocyte-specific depletion of aromatase in adult mice. 

It should be noted that TMX, which is used to induce the knockout of aromatase in the astrocytes of GFAP-ARO-iKO mice, is itself an estrogen receptor antagonist. We do not believe that TMX provided any confound in the current study for the following reasons. First, previous work has shown that TMX and its metabolites are completely cleared from the mouse brain and bloodstream within 7 days after the TMX injection [36]. To completely ensure no confounds from the TMX induction paradigm, we used a 3-week washout period in our studies. Secondly, other investigators have also successfully used TMX induction in conditional mouse knockouts to study the estrogen signaling and actions within the brain [37]. For instance, a TMX induction was successfully used to conditionally ablate the ER-α neurons in the brain of adult female mice, which helped to shed light on the role of ER-α in the estrogen negative feedback mechanism, in order to control gonadotropin secretion [37]. Finally, we also included several key controls in our study (GFAPCreERT2 mice, ARO-FLOX mice with TMX injections, and GFAP-ARO-iKO mice without TMX injections), which showed no significant effect of the TMX on any of the endpoints assessed in the control mice. Collectively, these findings indicate that TMX can be used successfully for knockout induction in estrogen studies, as long as requisite controls are included in the studies and there is an appropriate washout period to allow for the clearance of the TMX before the experiments are performed. 

An important finding was that the reactive astrogliosis in the hippocampus following the GCI was significantly reduced in the GFAP-ARO-iKO mice, although the microglial activation was increased. In addition, the GFAP-ARO-iKO animals experienced more neuronal death following the GCI compared to the control mice in the hippocampal CA1 region. These findings agree with the similar regulation of the reactive astrogliosis, microglial activation, and enhanced neuronal loss that was observed in the non-inducible GFAP-ARO-KO mice following the GCI [21]. Collectively, the results suggest that ADE_2_ acts to enhance this reactive astrogliosis while inhibiting the microglial activation following a GCI. 

The observed effects on the astrogliosis of the GFAP-ARO-iKO mice may, in part, underlie the neuroprotective effects of ADE_2_ observed in this and our previous study [21]. In support of this contention, previous work by others has demonstrated that GFAP/vimentin knockout mice exhibit reduced reactive astrogliosis and significantly enhanced neuronal damage following a cerebral ischemia [38], suggesting that reactive astrogliosis is neuroprotective after a cerebral ischemia. Furthermore, increased microglia activation after a cerebral ischemia has been shown to be associated with greater ischemic neuronal damage [39], and the ablation of microglia has been shown to reduce this ischemic damage following a cerebral ischemia [40]. Based on these findings, it is suggested that the effects of ADE_2_ in enhancing reactive astrogliosis and restraining microglia activation after a GCI may contribute to its neuroprotective effects in the brain following an ischemic injury. The direct protective effects of ADE_2_ on neurons could also contribute to its neuroprotective actions, as hippocampal neurons express all three estrogen receptor subtypes [41,42] and the direct protective effects of E_2_ on neurons in vitro has been demonstrated in many studies [43,44].

A limitation of this work is that we only examined one stressor (GCI) in our studies; thus, it is not clear whether the results can be extrapolated to other types of stressors, such as traumatic brain injury (TBI) (both penetrating and non-penetrating), or to focal cerebral ischemia (FCI). Compared to TBIs and FCIs, GCIs are unique, as they primarily involve a delayed cell death that occurs days after the injury and is especially noted in the highly vulnerable hippocampal CA1 region [45,46]. Thus, the pathological processes and temporal events between these different types of stressors can differ significantly. Nevertheless, aromatase expression has been shown to be increased in the brain after a penetrating and traumatic brain injury, as well as after an FCI [8,29,47,48]; thus, the E_2_ from astrocytes or neurons could contribute to neuroprotection for these other types of brain injuries/ischemias. Indeed, the administration of aromatase inhibitors suggests that brain-derived E_2_ is anti-inflammatory and neuroprotective in both penetrating and traumatic brain injuries [8,29,47,48], and global aromatase knockout mice and aromatase inhibitor studies following an FCI have shown a similar protective beneficial role for this brain-derived E_2_ [33]. To further elucidate the specific roles and functions of ADE_2_ and NDE_2_ in these different types of brain injuries/ischemias, it would be informative in the future to utilize GFAP-ARO-iKO mice and neuron-specific aromatase KO mice following a penetrating /traumatic brain injury or FCI. 

## 5. Conclusions

In conclusion, the current study provides a new inducible mouse model for the depletion of aromatase in the astrocytes of the adult brain. It also confirms a key role for ADE_2_ in regulating gliosis and neuroprotection following a GCI. This new GFAP-ARO-iKO mouse model should be very useful for the field in studying the various roles, functions, and mechanisms of ADE_2_ in the injured brain. 

## Figures and Tables

**Figure 1 biology-12-00621-f001:**
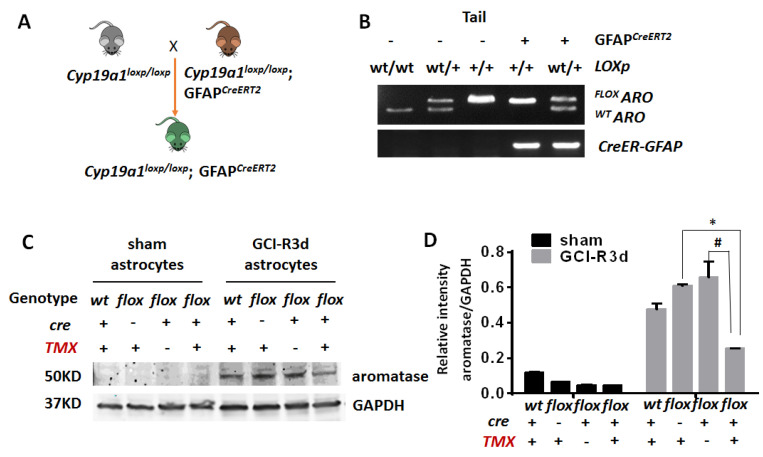
Creation and assessment of GFAP-ARO-iKO mice. (**A**). Breeding strategy represented in a diagram. (**B**). Tail DNA PCR confirmation of bitransgenic GFAP-ARO-iKO mice expressing GFAP-CreER and homozygous loxp*CYP19a1*. (**C**,**D**). Verification of the knockout of aromatase in astrocytes in the *Cyp19a1^loxp/loxp^*; GFAP*^CreERT2^* mice. Western blot (**C**) on purified astrocytes lysate isolated from sham or GCI-R3d mouse hemisphere. Densitometry analysis (**D**) demonstrates that the TMX-induced aromatase knockout animals at R3d showed a considerable reduction in aromatase expression of the astrocytes (**D**). n = 6–8 animals/groups. # *p* < 0.05 compared to TMX (−), and * *p* < 0.05 compared to cre (−), one-way ANOVA followed by post hoc.

**Figure 2 biology-12-00621-f002:**
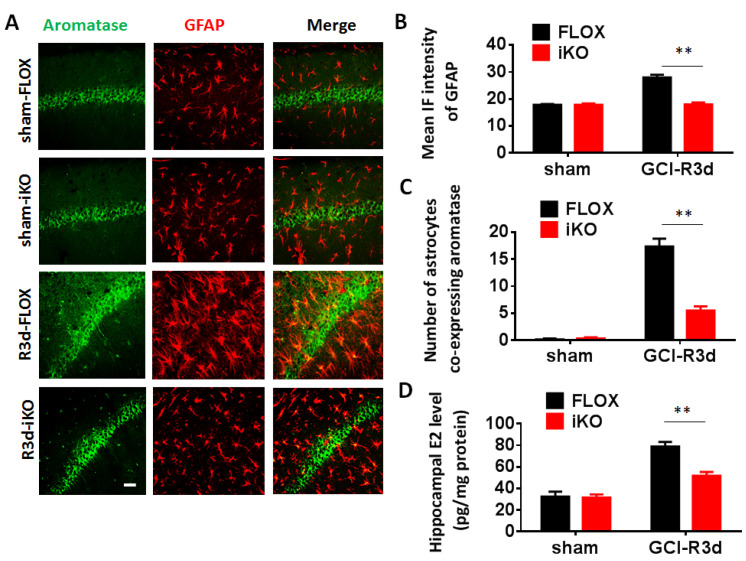
GFAP-ARO-iKO mice show significant diminishment of ischemia-induced aromatase expression in astrocytes, as well as reduced reactive astrocyte induction and hippocampal E_2_ elevation after GCI. (**A**). Representative images for aromatase and GFAP immunostaining in the hippocampal CA1. (**B**). Quantification of GFAP immunofluorescent intensity in the sham or R3d hippocampal CA1 regions of male FLOX and iKO mice. (**C**). Quantification for the number of astrocytes expressing aromatase in the hippocampal CA1 regions of sham and R3d groups. (**D**). Hippocampal E2 levels were measured by ELISA assay on hippocampal tissues from sham and R3d FLOX and iKO mice in male animals. n = 6–8 animals/group. Scale bars: 20 μm. ** *p* < 0.01 compared to FLOX, two-way ANOVA followed by post hoc.

**Figure 3 biology-12-00621-f003:**
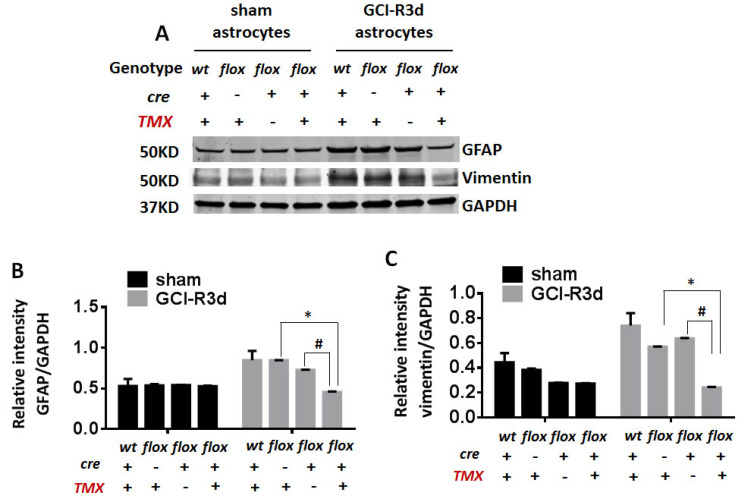
Decreased reactive astrogliosis in GFAP-ARO-iKO mice after GCI. Western blot analyses (**A**) on purified astrocytes lysate isolated from sham or GCI-R3d male mouse hemisphere and densitometry analysis demonstrated significant decrease in GFAP (**B**) and vimentin (**C**) expression in the astrocytes in GFAP-ARO-iKO mice at R3d. n = 6–8 animals/group. # *p* < 0.05 compared to TMX (−), and * *p* < 0.05 compared to cre (−), one-way ANOVA followed by post hoc.

**Figure 4 biology-12-00621-f004:**
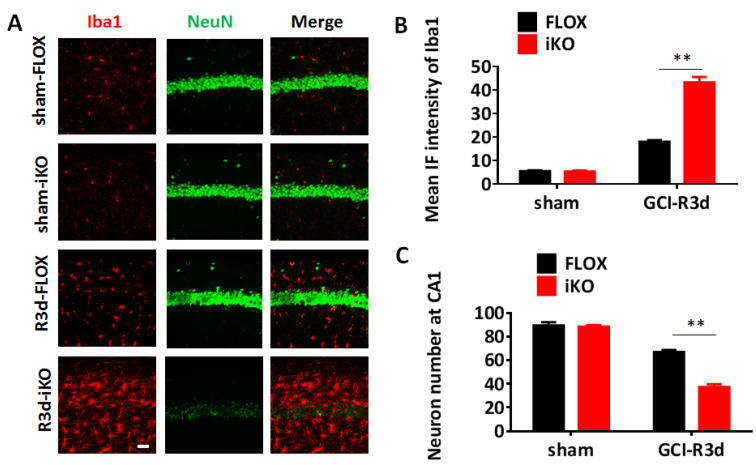
GFAP-ARO-iKO mice exhibit enhanced neuronal loss and microglial activation after GCI. (**A**). Representative images of NeuN and Iba1 immunostaining in the sham or R3d hippocampus and CA1 regions of male FLOX and GFAP-ARO-iKO mice. (**B**). Quantification of mean immunofluorescence (IF) intensity of Iba1. (**C**). Quantification of CA1 neuron number by NeuN staining. Scale bars: 20 μm. n = 6–8 animals/group. ** *p* < 0.01 compared to FLOX, two-way ANOVA followed by post hoc.

## Data Availability

No new data were created or analyzed in this study. Data sharing is not applicable to this article.

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
