# Peer review of "Development and Characterization of Inducible Astrocyte-Specific Aromatase Knockout Mice"

_biology, 2023, doi:10.3390/biology12040621_

Round 1

Reviewer 1 Report

In this manuscript, the authors described the development and characterization of inducible astrocyte-specific aromatase knockout mice. They generated a tamoxifen inducible astrocyte-specific aromatase knockout (GFAP-ARO-iKO) mouse model, and performed experiments demonstrating that GFAP-ARO-iKO mouse brain shows a significant depletion of aromatase expression in astrocytes, as well as hippocampal E2 level after GCI. Interestingly, the GFAP-ARO-iKO mice also have decreased astrogliosis, increased neuronal damage and microglial activation after GCI. Thus, the authors concluded that the GFAP-ARO-iKO mouse model provide an important new model to help elucidate the roles of astrocyte-derived estrogen in the brain. The manuscript is well-written, and the experiment data is well presented. There is a few minor points need to be addressed:

1.          In the Methods “induction of aromatase deletion in GFAP-ARO-iKO” mice, the authors indicate that TMX was injection 3-5 consecutive days. The authors should provide detailed information on how many TMX injections were given for each experiment.

2.          For the GCI surgery, the authors should provide information on the exclusion criteria of the animal, and indicate the successful rate of the surgery.

3.         Figure 2, the authors should present the quantification of Aromatase co-localization with GFAP.

Reviewer 2 Report

The authors generated a tamoxifen (TMX)- inducible astrocyte-specific aromatase knockout mouse line to study astrocytic E2 effects in adult mice after development. They did not observe changes in astrocyte morphology and distribution, and the brain anatomy was normal in these mice. Following brain injury, KO mice showed defective reactive astrogliosis, increase in neuronal loss and microglia activation. The results are consistent with the published data of non-inducible aromatose KO mice and appropriate control mouse lines were used to confirm the interpretation of results.

The studies are well performed with appropriate controls. The manuscript is well written.

There are only minor points to be addressed:

- Figure legends should be revised to systematic labeling of panels.  Panels C and D of Figure 1 are a bit confusing; “C and D” shall be removed and D shown in bold. Size of the scale bar is missing.

- In Figure 1 panel D the text “Reletive” have to be changed to “Relative”. 

- line 54, remove extra dot.

Reviewer 3 Report

Review of the manuscript entitled: Development and Characterization of Inducible Astrocyte-Specific Aromatase Knockout Mice.

In my opinion, the manuscript was very carefully prepared, does not require major corrections, but some corrections should be made. In abstract and introduction clear aim of the manuscript should be added e.g. "The aim of the present study was to ...".

Method section - the number of animals (even an estimate) should be added.

All abbreviations should be explained e.g line – RIPA, and so on.
